# Wood Surface Modification with Hybrid Materials Based on Multi-Walled Carbon Nanotubes

**DOI:** 10.3390/nano12121990

**Published:** 2022-06-09

**Authors:** Madalina Elena David, Rodica-Mariana Ion, Ramona Marina Grigorescu, Lorena Iancu, Mariana Constantin, Raluca Maria Stirbescu, Anca Irina Gheboianu

**Affiliations:** 1National Institute for Research & Development in Chemistry and Petrochemistry—ICECHIM, 060021 Bucharest, Romania; rmgrigorescu@gmail.com (R.M.G.); lorenna77ro@yahoo.com (L.I.); marriconstantin@yahoo.com (M.C.); 2Doctoral School of Materials Engineering Department, Valahia University of Targoviste, 130004 Targoviste, Romania; 3Institute of Multidisciplinary Research for Science and Technology, Valahia University of Targoviste, 130004 Targoviste, Romania; stirbescu_nic@yahoo.com (R.M.S.); anca@icstm.ro (A.I.G.)

**Keywords:** multi-walled carbon nanotubes, nanoparticles, sound oak wood preservation, hybrid nanomaterials, surface modification

## Abstract

In this work, new treatments based on multi-walled carbon nanotubes (MWCNTs), MWCNTs decorated with zinc oxide (ZnO), MWCNTs decorated with hydroxyapatite (HAp) and MWCNTs decorated with silver (Ag) nanoparticles dispersed in PHBHV solution are proposed for improving sound oak wood properties. We hypothesize that the solutions containing decorated MWCNTs will be more efficient as wood consolidants, not only because of the improved mechanical properties of the treated wood but also because of the hydrophobic layer created on the wood surface. In order to test these hypotheses, the treatments’ potential was investigated by a number of complex methods, such as colorimetric parameter measurements, water absorption tests, mechanical tests, artificial aging and antifungal tests. The data confirm that the treated wood materials have moderate stability, and the color differences are not perceived with the naked eye. A significant improvement of the treated samples was observed by water absorption, humidity and mechanical tests compared to untreated wood. The best results were obtained for samples treated by brushing with solutions based on decorated CNTs, which confirms that a uniform and thicker layer is needed on the surface to ensure better protection. The wood behavior with accelerated aging revealed that the control sample degraded faster compared to the other treated samples. Antifungal tests showed that higher growth inhibition was obtained for samples treated with 0.2% MWCNTs_ZnO + PHBHV. Considering all of the obtained results, it can be concluded that the most effective treatment was MWCNTs_ZnO + PHBHV at a nanocomposite concentration of 0.2%, applied by brushing. Thus, wood protection against mold and fungi will be achieved, simultaneously ensuring improved mechanical strength and water barrier properties and therefore maintaining the structural integrity of sound oak wood over time.

## 1. Introduction

Over time, wooden artifacts are constantly subjected to several degradation factors, which more or less affect the structural integrity and mechanical strength of these materials [1,2]. The main factors that affect their integrity are: high relative humidity, extreme temperatures and biological attack by insects, fungi and microorganisms [3,4,5].

Therefore, wood consolidation has become a serious problem, and researchers are trying to obtain new systems in order to find the most efficient and convenient consolidation solution that ensures better durability over time. The coating applied to the wood surface must be able to prevent or limit the deteriorating effects of environmental factors and to ensure that the aesthetic parameters of the wood are maintained [1,6,7].

Research in the field has shown that wood modification can improve the dimensional stability, durability, strength and other properties of wood. Nanotechnology development has provided more possibilities for wood modification, and one of the preferred methods entails the chemical modification of wood. In the last several years, nanomaterials have played a key role in the performance of wood preservatives due to their unique characteristics that enhance the wood properties [6]. For example, superhydrophobic surfaces with antimicrobial properties have been achieved using nanoparticles (NPs), such as zinc oxide (ZnO) NPs [8,9]. Favarim H.R. and Leite L.O. investigated the performance of pine wood impregnated with ZnO NP solution regarding the resistance to fire and ultraviolet radiation. The results showed an important improvement in fire retardancy for all of the treated samples due to the effectiveness of the nanomaterial as a thermal-protective barrier [10]. Clausen C.A. and co-workers treated southern yellow pine and yellow poplar wood with ZnO NPs in order to evaluate the termite resistance. The treated material showed termite feeding inhibition with up to 35% mortality, which suggests that the used treatment has desirable wood protection properties [11]. In another study, beech wood blocks were treated with ZnO NPs by using a modified dip method in order to investigate the wood’s water absorption properties. The results indicate that the treatment greatly improved the dimensional stability and reduced the hygroscopicity of the treated wood [12]. In addition, it was reported that the presence of ZnO NPs on the wood surface led to increased mechanical properties in comparison to control specimens [13,14]. In another study, it was reported that the presence of ZnO NPs on the surface of wood enhanced the hydrophobic properties and anti-UV photochromic properties [15].

Silver (Ag) NPs are another type of nanomaterial with high potential in wood protection. Over the years, Ag NPs have proven their usability in various domains, including as consolidants on different artifacts and works of art. The main advantages of these nanoparticles, as compared to the other noble metals, are: non-toxicity, stability under environmental conditions, wide-range absorption of light and antimicrobial activity [1,16]. The environmental impact of a wooden material treated with Ag NPs was studied by T. Künniger and co-workers, and it was established that the metallic Ag NPs transformed into Ag complexes that are less toxic than ionic Ag [17]. The fungicide effects of Ag NPs on wood were demonstrated by Akhtari M. and co-workers [18]. In another study, commercial timbers were treated with Ag NPs in order to test their effect on white-rot (*Trametes versicolor*) and brown-rot (*Lenzites acuta*). A lower weight loss was observed for the treated wood compared to the control for both species of fungi [19]. Additionally, the decay resistance of the pine wood was increased by impregnating the wooden material with an autoxidized soybean oil polymer containing Ag NPs [20]. Another type of nanomaterial with potential for wood preservation is hydroxyapatite (HAp). This nanomaterial has received significant attention as a consolidant for various types of materials. For example, Ion R. and co-workers treated young and aged wood with HAp in order to investigate its consolidating efficiency. It was reported that the presence of HAp NPs on the wood surface significantly improved the mechanical properties (from 29 MPa for untreated wood to 35.8 for treated wood) [21]. 

Nanotubes are a new class of materials with promising properties as coatings on wood surfaces. These materials present the advantage that nanoparticles or biocides can be loaded on their hollow structures, thus obtaining a new nanomaterial with different properties from those of the initial nanotube [6,22]. To date, CNTs have been used in a few studies as coatings for wood preservation. In a study, CNTs were embedded in epoxy resin and used as a coating for timber structures. It was demonstrated that this coating significantly improved the mechanical properties of wood compared with the control [23]. In another study, it was reported that the flexural and tensile properties of wood–polymer nanocomposites increased with the increasing content of multi-walled carbon nanotubes (MWCNTs) [24]. In addition, Zhang Y. and co-workers reported that the addition of a small amount of CNTs can significantly increase the mechanical properties (tensile strength, bending strength and elasticity) of the wood/polyether sulfone composite material [25].

In this study, new materials based on MWCNTs and MWCNTs decorated with ZnO NPs, Ag NPs and HAp NPs obtained by our team [26] were dispersed in a solution of poly(3-hydroxybutyrate-co-3-hydroxyvalerate) (PHBHV) in order to study their efficiency as a consolidation solution for sound oak wood. We hypothesized that the obtained solutions containing NPs on the CNT surfaces would be more suitable as consolidant treatments not only because of the improved mechanical properties of the treated wood but also because of the hydrophobic layer created on the wood surface. Furthermore, we believed that the treatment methods and nanocomposite concentrations would influence the properties of the wood. In order to test these hypotheses, the treatments’ potential was investigated by a number of complex methods, such as consolidant retention, colorimetric parameters, water absorption, humidity, contact angle, mechanical properties, artificial aging and antifungal tests. Sound oak wood was chosen as the substrate because it is considered to be one of the most used types of wood for building houses, gates, churches, furniture and household items.

## 2. Materials and Methods

In this study, the tubular nanomaterials obtained in our previous studies [26,27] were used. Each type of nanocomposite (MWCNTs, MWCNTs_ZnO NPs, MWCNT_Ag NPs and MWCNTs_HAp NPs) was dispersed at different concentrations (0.1, 0.2 and 0.4%) in a solution of 2% PHBHV in chloroform in order to obtain a better dispersion of the nanomaterials.

### 2.1. Nanomaterial Dispersion in PHBHV Solution

Nanocomposite dispersions based on MWCNTs were fabricated in a reflux plant, where, in the first step, a polymer solution was obtained by dissolving 1.6 g of PHBHV (molecular weight of 67,000 g/mole containing 2% hydroxyvalerate—Good Fellow) in 70 mL of chloroform at 60 °C for 6 h. In the second step, a solution of MWCNTs and 10 mL of chloroform was obtained by sonication for 1 h at room temperature, and then this solution was added to the polymer solution. Subsequently, the resulting mixture was sonicated for 30 min at room temperature in order to obtain MWCNTs_PHBHV. PHBHV was chosen because, compared to other types of polymers, it has attracted attention as a new sustainable biopolymer due to its low environmental impact (biodegradability and non-toxicity) [28,29]. Other advantages include the fact that the presence of this type of polymer leads to improvements from the point of view of the mechanical and hydrophobic properties of the system [30].

### 2.2. Wood Preparation

The wooden material used in this study was young oak wood (Quercus robur). Sound oak wood was dimensioned (25 × 23 × 8 mm^3^) and conditioned for one month at a temperature T = 20 ± 3 °C and a relative atmospheric humidity φ = 55 ± 5%. After the consolidation products were obtained, the pieces were treated using three different techniques (brushing—in three layers; spraying—in three layers; and total immersion—in solution for 15 min). Subsequently, the samples were allowed to dry at room temperature and were reconditioned for another month. After that, the treated samples and control samples were used in order to investigate the treatment applicability.

### 2.3. Consolidant Retention (CR)

This factor was investigated by a gravimetric method, i.e., measuring the mass percentage increase (Equation (1) [31]), by treating wood samples with each composition and with each method of treatment application (brushing, spraying and immersion). The degree of impregnation of the wooden support was monitored according to the method of application of the treatment in order to evaluate the effectiveness of the treatment. The experiment was implemented in triplicate.
(1)$$\mathrm{CR}=\frac{\left(m_{f}-m_{i}\right)}{m_{i}}*100$$
where CR is the mass percentage increase in [%]; m_f_ is the final mass of the treated and conditioned sample in [g]; and m_i_ is the initial mass of the conditioned sample in [g].

### 2.4. Optical Microscopy (OM)

In this study, optical microscopy was used to study the uniformity of the treatment applied to the wood surface and the thickness of the layer obtained depending on the application technique. These characteristics were investigated using the NovexMicroscope BBS, which offers the possibility to investigate the samples in transmitted light, with magnification between 4–100×. The equipment has a digital video camera (EUROMEX, Arnhem, The Netherlands) attached, which, through the microscope software (ZenPro), allowed the acquisition of data in real time.

### 2.5. Fourier Transform Infrared Spectroscopy (FTIR)

The IR spectra were recorded using a GX-type spectrometer (Perkin Elmer, Waltham, MA, USA), which allows measurements in the range of 4000–600 cm^−1^. The recording of the spectrum collection was performed in total attenuation reflection mode at a resolution of 4 cm^−1^ for the accumulation and mediation of 32 spectra.

### 2.6. Wavelength-Dispersive X-ray Fluorescence (WDXRF)

WDXRF was performed in order to determine the qualitative and quantitative elementary composition of the samples. The device is equipped with 3 crystal analyzers (with automated exchange): LiF (200) for heavy elements (Ti-U), PET and RX 25 for light elements (O-Mg and Al-Sc) at 200 W power (50 kV tens, 4 mA int). Detection limit: 1 ppm–10 ppb; accuracy < 0.1–0.5%; the elements ranged from 8 O to 92 U.

### 2.7. Colorimetric Tests

Chromatic parameters were investigated in order to determine the treatment influence on the natural color of the wood. Thus, a CR-410 colorimeter (Konica Minolta, Tokyo, Japan) set in the CIE L*a*b* system (CIE 1986) was used. Three determinations were performed for each sample, before and after the application of the treatment, and the arithmetic mean was calculated for each sample. The total color difference (ΔE_xfinal_) was calculated according to [32] using Equation (2):(2)$${\Delta E}_{x\;\mathrm{final}}={\left({\Delta L}_{x}^{2}+{\Delta a}_{x}^{2}+{\Delta b}_{x}^{2}\right)}^{1/2}$$
where ΔL is the difference in lightness, calculated with the formula: ΔL = L_treated sample_ − L_untreated sample_; Δa is the chromatic deviation of the coordinates of a* coordinates, calculated with the formula: Δa = a_treated sample_ − a_untreated sample_; and Δb is the chromatic deviation of the b* coordinates, calculated with the formula: Δb = b_treated sample_ − b_untreated sample_.

### 2.8. Water Absorption Test (WA)

The water absorption test involved drying the samples in an oven at 103 °C. After drying, the samples were allowed to cool at room temperature and were then weighed (W_1_). Subsequently, the samples were immersed in distilled water for 24 h at room temperature, and after 24 h, the samples were removed from water, wiped with a towel and weighed (W_2_). WA was calculated according to [33] using Equation (3). The determinations were performed in triplicate for each sample, and the arithmetic mean was calculated.
(3)$$\mathrm{WA}=\frac{W_{2}-W_{1}}{W_{1}}*100$$

### 2.9. Humidity Test (U)

Another test to investigate the degree of water absorption of the samples exposed to certain environmental conditions was the U test. This test involved immersing the wood samples in distilled water for 30 min and then weighing them (W_i_). The samples were then dried in the oven for 1 h at 100 °C and left for 24 h at room temperature, according to ISO 22157-1: 2004. The determinations were performed in triplicate for each sample, and the arithmetic mean was calculated. Subsequently, the samples were weighed (W_f_), and U was calculated according to [34] using Equation (4).
(4)$$U=\frac{W_{i}-W_{f}}{W_{f}}*100$$

### 2.10. Contact Angle

This test was carried out in order to investigate the hydrophobicity of the wood surface before and after treatment. The sample was placed on a straight surface near a light source, and 6 μL was dropped at a single point on the surface of the sample using a micropipette. After the drop reached the surface, photographs were taken at a distance of 10 cm from the sample, every 15 s for 1 min, in order to calculate the contact angle at different times. The measurements were recorded at a temperature of 23 ± 1 °C. The contact angle was calculated using the DropAnalysis plugin LB-ADSA from ImageJ according to [35]. The determinations were performed in duplicate for each sample, and the arithmetic mean was calculated.

### 2.11. Mechanical Tests

Measurements of the mechanical strength, expressed by determining the rebound number for untreated and treated samples, were recorded with a Silver Schmidt Proceq hammer, type L, with an impact energy of 0.735 Nm [36]. For each sample, 10 measurements were performed, with a minimum edge distance of 5 mm. The hammer was positioned at 90° on the sample. The compressive strength, expressed in MPa, was calculated according to Equation (5) using the arithmetic values of the rebound number, determined in duplicate:(5)$$\mathrm{Compressive}\;\mathrm{strength}=2.77{\;*\;e}^{0.048*Q}$$
where 2.77 is the device constant, and Q is the rebound number.

### 2.12. Accelerated Aging Behavior

The main factors that influence the aging process of wooden materials in nature are UV radiation, humidity and temperature. In this study, the testing of consolidant efficiency was achieved under simulated environmental conditions.

#### 2.12.1. Artificial Aging Test by UV Radiation Exposure

This study was carried out in order to investigate the effect of UV radiation on wood surfaces for 400 h, the equivalent of two years of aging in natural conditions [37]. UV irradiation was performed using an 18 W UV bulb lamp. Wooden samples were positioned on a support at a distance of about 10–20 cm from the lamp and an inclination of 45 °C, according to [38]. Under these conditions, the samples had a temperature of 24 ± 2° C and a relative humidity of 52 ± 5%. Colorimetric parameters were measured after 100, 200, 300 and 400 h to investigate color changes for each wood sample. The determinations were performed in duplicate for each sample, and the arithmetic mean was calculated.

#### 2.12.2. Artificial Aging Test by Exposure to Temperature Variations

An artificial aging test involving exposure to temperature variations was carried out according to [39], and colorimetric parameters were recorded before and after aging. This test consisted of 20 alternative cycles, each containing three steps: step I consisted of drying the wood samples for one hour at a temperature of 50 °C; in step II, the samples were conditioned in a laboratory environment for 1 h (T = 25 ± 2 °C), and step III consisted of keeping the samples in the freezer at a temperature of −20 ± 2 °C. The determinations were performed in duplicate for each sample, and the arithmetic mean was calculated.

### 2.13. Antifungal Activity

Untreated and treated wood samples were embedded in a specific culture medium for the growth and isolation of fungi (*Aspergillus niger* and *Penicillium* sp., in vitro) individually per sample. Solid Sabouraud medium was prepared according to the protocol in [40] and was used as a culture medium for growing and isolating fungi. Additionally, as the inoculum, a suspension of sterile physiological water with a concentration of 1–3 × 106 spores/mL from a fresh fungal culture grown for 4 days on solid PDA medium (Scharlau, peptone—4; glucose—20; agar—15 (g/L)) was used. Petri dishes with sterile Sabouraud medium were seeded in cloth with a sterile swab. Each wood sample was placed in the center of the Petri dish, and the Petri dishes were incubated for up to 5 days at 28 °C. During this time, the plates were observed and photographed to visually assess the absence or presence of growth of *Aspergillus niger* or *Penicillium* sp. on the surfaces of the wood samples.

### 2.14. Statistical Analysis

The colorimetric and mechanical results were studied by two-factor ANOVA. Statistical analyses were performed using Microsoft® Excel®, version 2010 14.0 (Redmond, WA, USA). The obtained results were compared by Tukey’s test (*p* < 0.05).

## 3. Results

### 3.1. Consolidant Retention

The CR on the surface of the material is the most important aspect in the consolidation process, because this aspect leads to the improvement of the material properties. Thus, after the material was treated with the consolidation solution, the conditioning phase was carried out, where the solvent evaporates, and the solid consolidant was fixed in the structure of the material, resulting in weight gain—CR. The amount of consolidant retained in the wood structure was calculated for each sample. The highest CR was obtained by brushing (Figure 1), while low values of CR were obtained for the other two investigated methods. This suggests that the applied method can significantly improve the coating homogeneity and thus the wood properties by offering better and stronger protection of the surface.

### 3.2. Optical Microscopy

OM images revealed that all of the used treatments were uniformly distributed on the surface of the wood samples. As can be observed (Table 1), in the case of samples treated by brushing, a slight agglomeration of particles occurred, and a thicker layer of polymer (of about 6 µm) was found on the surface of the wood treated by brushing, which leads to greater protection. Thinner layers of polymer were obtained in the case of the other two methods, and in the case of the spraying method, there was also a slight reduction in the size of the nanomaterials. The nanocomposite size reduction in this case was due to the spraying method. These results are in agreement with the results obtained by the CR method.

### 3.3. FTIR Analysis

The FTIR spectra recorded for untreated and treated samples are presented in Figure 2. The spectrum obtained for untreated wood confirms the presence of lignin and holocellulose (hemicellulose and cellulose) due to the wide region from 3300–3600 cm^−1^ (specific to the vibration of OH stretching and poor water absorption) and the band from 2900 cm^−1^ (C-H stretching vibration of methyl and methylene groups). The main characteristic bands of lignin are present at 1596, 1507, 1459 and 1417 cm^−1^, corresponding to C-O deformation vibrations of the guaiac-type ring; the band at 1314 cm^−1^ corresponds to the CH group, and the band at 1224 cm^−1^ corresponds to tensile vibrations of C = C in aromatic rings [41]. The characteristic hemicellulose band is observed at 1730 cm^−1^ and attributed to the C = O stretching vibration of acetyl, carboxyl and carbonyl groups (characteristic of deciduous wood species) [42]. The deformation vibration of CH_2_ in cellulose and hemicellulose is observed at 1369 cm^−1^, and the band at 1015 cm^−1^ is characteristic of the C−O−C stretching vibration of the primary alcohol in cellulose and hemicellulose [41]. After wood treatment with the consolidation solutions, new functional groups were observed in the FTIR spectra. In the case of the wood treated with the solution based on MWCNTs + PHBHV, the characteristic bands of MWCNTs are observed at 2846 and 1374 cm^−1^, attributed to C-H stretching vibration, with CH_3_ stretching vibration at 1426 cm^−1^ [26]. The polymer presence was confirmed by its characteristic bands from 800–1200 cm^−1^, attributed to the tensile vibration of C = O of the ester group [43].

In the case of the solution based on MWCNT_ZnO dispersed in PHBHV, the addition of ZnO is confirmed by both the band at 981 cm^−1^, characteristic of the Zn-C group, and that at 517 cm^−1^, attributed to the Zn-O group. The FTIR spectrum of wood treated with MWCNT_HAp + PHBHV solution confirms the presence of major functional phosphate groups (P-O) at 1028 cm^−1^ and 888 cm^−1^. The band at 1588 cm^−1^ is proof of the presence of the carbonate functional group. In the case of the solution based on MWCNT_Ag + PHBHV, the presence of small peaks between 627 and 481 cm^−1^, corresponding to the deformation vibrations of the Ag-O bond, is observed [26].

### 3.4. WDXRF Analysis

The composition of untreated samples and samples treated with different consolidation solutions was determined by WDXRF analysis. The analysis of oxides is presented in Table 2 and confirms the presence of the top components of the wood structure, such as calcium, iron, silicon, sulfur or potassium. Additionally, after wood treatment with consolidation solutions, the presence of new oxides is observed in small proportions, specific to each solution.

### 3.5. Colorimetric Tests

Colorimetric parameters are important because the applied treatment must not significantly change the original color of the material. This technique was used to investigate chromatic parameters before and after the application of consolidation treatments. The difference in lightness (ΔL_x_) for control and wood materials treated with solutions of different concentrations is shown in Figure 3. After treatment application, low lightness is observed for most of the samples compared to the wood before treatment due to the slightly closed color of the treatment. In the case of treatment based on MWCNTs_Ag + PHBHV, an increase in lightness is observed at lower concentrations. This occurs because the treatment has a lighter color compared to the other applied treatments. Overall, it can be concluded that the luminosity of the samples is not significantly changed after the treatments.

In the yellow–blue range (Δb_x_), some small color changes can be observed after the treatment application (Figure 4). For most treated samples, Δb_x_ is less than 3, which confirms that after the treatment, the wooden materials have moderate color stability [44], and the color changes are difficult to detect with the naked eye.

The total color difference (ΔE_x_) calculated according to Equation (2) confirmed that the natural color of the wood did not significantly change after the treatment application (Figure 5). It can be observed that at lower concentrations of nanocomposites, the values are lower, less than 2.5, and when the concentration of nanocomposites in solution increases up to 0.4%, the values easily increase, but the modifications are still in the acceptable range (below 5) according to ASTM 2244 [32]. This means that the higher the value of ΔE_x_, the farther the color is from the original shade. The colorimetric standard suggests that the perfect color compared to the original one presents ΔE_x_ = 0, and it cannot be detected by the human eye. The minimum difference detectable by the human eye is ΔE_x_ = 2.5. A ΔE_x_ value between 3 and 5 is considered an acceptable number, but the color difference can be perceived by the human eye. ΔE_x_ values above 5 are considered unacceptable because the original color of the material changes significantly. As can be observed, ΔE_x_ values between 3 and 5 were obtained when a concentration of nanocomposites of 0.4% was used, suggesting that the quantity of the nanocomposites in the solutions must be limited.

#### Data Analysis

According to Figure 5, we started from the hypothesis that the natural color of the wood did not significantly change after the treatment application. An analysis of variance (ANOVA) was performed, considering the total color difference after the sample treatment to be a dependent factor (Table 3). Treatments were established by the interaction between factors, that is, three concentrations (0.1, 0.2 and 0.4%) and three methods (brushing, spraying and immersion). According to the p-values, the wood pieces treated with MWCNT_ZnO + PHBHV and MWCNT_HAp + PHBHV solutions presented p-values less than 0.05, which means that there is a low probability of obtaining that result by chance when the treatment has no real effect. In these cases, neither the concentration nor the applied method significantly changed the natural color of the wood. On the other hand, in the case of wood pieces treated with MWCNTs + PHBHV and MWCNTs_Ag + PHBHV solutions, it can be observed that a lower probability of obtaining that result by chance when the treatment has no real effect was obtained for the method factor. Thus, the applied method plays a key role in obtaining a consolidation treatment with high-performance properties. Additionally, the analysis of variance confirmed that a lower p-value was obtained for cases in which the treatments were applied by brushing at a concentration of 0.2% nanocomposites. The obtained results support our hypothesis that the composition of the consolidation solutions, the nanocomposite concentration and the method of the solution application will significantly enhance the wood properties.

### 3.6. Water Absorption Test

In order to evaluate the effectiveness of the obtained treatments, the WA capacity of the untreated and treated wood samples was investigated (Figure 6). One of the most important characteristics of wooden materials is their behavior in a humid environment. An effective reinforcement treatment must protect the wood against moisture, especially in the case of aged wood, because this type of wood is much more sensitive to humidity variations compared to young wood [1]. The WA test revealed that all of the treated samples showed lower water absorption compared to the control sample. Similar values were obtained for all treated samples, the main reason being that the same polymer concentration was used for all of the applied treatments. In general, wood has an increased water absorption capacity due to its chemical composition (wood fiber contains cellulose and hemicellulose, components that contain numerous hydroxyl groups) [45], so a treatment that limits water penetration into the wood is essential. Moreover, in all of the presented cases, it can be observed that a lower water amount was found for the samples treated by brushing.

### 3.7. Humidity Test

The results obtained after the humidity tests support the effectiveness of the used treatments, the best results being obtained for the materials treated by brushing and spraying (Figure 7). Therefore, the results obtained from the WA test and the U test confirmed that the treatments used in this study successfully created a hydrophobic surface, which has the role of protecting the wood surface against long-term exposure to humidity and to environmental changes, respectively.

### 3.8. Contact Angle

The hydrophobic properties of the obtained compositions were investigated by contact angle measurements. The higher the value of the contact angle, the lower the probability that water can penetrate the wood. According to the obtained results (Figure 8), it can be seen that, after applying the treatment to the wood samples, the hydrophobicity greatly improved compared to the untreated samples. The highest values of the contact angle were obtained for samples treated with solutions of 0.2 and 0.4% concentrations applied by brushing and spraying. This suggests that applying the treatment using these methods leads to a much more uniform layer on the wood surface, thus contributing to higher hydrophobicity.

### 3.9. Mechanical Tests

Mechanical tests confirmed that the presence of the obtained solutions significantly improved the compressive strength of the treated wood (Figure 9). Additionally, improved mechanical properties were observed when solutions based on decorated nanotubes were used, which means that the presence of nanoparticles makes a contribution in this regard. This aspect is also supported by literature papers, confirming that the presence of nanoparticles significantly improves the mechanical properties of polymeric films [46,47,48]. Another aspect observed was that the application method of the consolidation solutions also had a major effect on the mechanical properties of the wood, as applying the solutions by brushing and spraying provided superior strength.

#### Data Analysis

Based on the obtained results from the mechanical test, we considered that the applied method and the modified nanotube treatments significantly improved the mechanical properties of the wood pieces. An analysis of variance was performed, considering the compressive strength obtained after the sample treatment to be a dependent factor (Table 4). Treatments were established by the interaction between factors, that is, three concentrations (0.1, 0.2 and 0.4%) and three methods (brushing, spraying and immersion). According to p-values, the wood pieces treated with MWCNTs_ZnO + PHBHV solutions presented p-values less than 0.05, which means that the results are statistically significant and the ZnO NPs have a major contribution to obtaining superior mechanical properties. In addition, the analysis suggested that the applied methods significantly improved the mechanical character of wood for all considered treatments. On the other hand, it can be observed that the nanocomposite concentration had no significant effect on these properties. Therefore, also in this case, the applied method plays a key role in obtaining a consolidation treatment with high-performance properties, because, depending on the method, the solution penetrates better into the wood pores.

### 3.10. Accelerated Aging Behavior

#### 3.10.1. Artificial Aging Test by UV Radiation Exposure

An artificial aging test using UV radiation exposure was performed in order to study the effect of radiation on wooden material and the effects of long-term aging. In Table 5, ΔE_x_ vs. exposure time is presented. It can be observed that most of the samples underwent color darkening in the first 100 h of exposure. However, these values are below the threshold of 2.5, which means that the changes are almost imperceptible to the naked eye. After 200 h of exposure, the control sample exceeded the 2.5 limit but was less than 5, which means that the changes are still in the acceptable range. For samples treated with consolidation solutions, a slight increase in values is observed, but the values remain in the visible range, which confirms the effectiveness of the treatments. After 300 h, the 5 limit (acceptable range) was slightly exceeded in the case of the wood samples treated with MWCNTs_PHBHV, which means that the solutions based on decorated CNTs are more stable, increase the stability of the polymer and offer greater protection. After 400 h, a significant increase in ΔE_x_ was observed for the control sample, with the wood color blackening. Regarding the treated samples, after 400 h of exposure, the acceptable limit was slightly exceeded in the case of solutions with 0.1 and 0.2% nanomaterial concentrations, except for the samples treated by brushing.

#### 3.10.2. Artificial Aging Test by Temperature Variation Exposure

The environmental conditions to which wood is continuously subjected (temperature or UV radiation) cause the discoloration of wooden materials, and this takes place in shades of grey, brown, yellow or red, depending on the type of wood [49]. The process of wood discoloration occurs due to environmental factors that play an important role in the degradation of lignin or hemicellulose in the cell wall of the wood material. After 20 cycles of temperature variations, the control sample presented the largest change in lightness, which showed a slight discoloration over time (Figure 10). Additionally, samples based on MWCNTs + PHBHV showed discoloration compared to the initial sample. The wood samples treated with decorated nanotubes presented higher color stability, which means that the presence of nanoparticles in the solution contributed to stability against temperature variations.

### 3.11. Antifungal Activity

The presence of bacteria and fungi in the wood structure leads to a concomitant interaction, which accelerates the disintegration of the wood material. The population of microorganisms varies depending on the state of wood degradation, and they develop more and more as the wood decomposes in different environmental conditions, mainly in the presence of water and oxygen. In addition, the development of fungi is favored when the humidity of the wood exceeds 20% [50]. Therefore, a conservation treatment is needed that not only strengthens the weakened wood but also prevents the deformation and decomposition of the wood. The antimicrobial properties of the obtained nanocomposites were already tested in our previous study [26], where it was demonstrated that MWCNTs_ZnO NPs and MWCNTs_Ag NPs presented better antimicrobial properties against Gram-positive (*S. aureus* and *B. subtilis*) and Gram-negative bacteria (*P. aeruginosa* and *E. coli*) and yeast (*C. albicans*). Figure 11 shows the fungal growth on untreated oak wood and oak wood treated for 72 h.

According to the obtained results, the solutions containing a 0.2% nanocomposite concentration applied by brushing were further selected in order to investigate the antifungal activity of the obtained coatings. This selection was based on the obtained results of the colorimetric test, as the natural color of the samples was not modified by using this concentration, and also other tests, such as mechanical, water absorption, humidity or accelerated aging behavior, which presented enhanced properties at this concentration applied by brushing. The fastest growth of *Aspergillus niger* was observed for the control samples, followed by woods treated with MWCNTs + PHBHV and MWCNTs_HAp + PHBHV. The slowest growth of fungi was observed in the case of wood treated with MWCNTs_ZnO + PHBHV and MWCNTs_Ag + PHBHV. The presence of Ag or ZnO nanoparticles considerably influenced the growth of fungi on the wood surface, because the treatments acted as a protective layer on the wood surface. In addition, in the case of *Penicillium* sp., the wood samples treated with MWCNTs_ZnO + PHBHV led to the inhibition of the development of the aerial mycelium. At 72 h, it was observed that the control sample was covered with sporulated mycelium.

## 4. Discussion

Reversibility is a mandatory requirement for materials used in heritage conservation, including hydrophobic protectives. Presently, current wood protectives are not actually reversible, and they remain on the wood surfaces for a long time after their hydrophobicity is lost and can hardly be removed. It can significantly affect the wood re-treatability and further conservation interventions. For this reason, we selected the copolymer PHBHV, because its main potential is ‘reversibility by biodegradation’ once water repellency ends. PHAs were proposed because of their intrinsic biodegradability in environmental conditions in order to generate temporary treatments that do not need any removal, which is an important target for the protection of cultural heritage [51]. Furthermore, it has been reported that PHBHV does not promote fungal colonization on wooden materials, improves the mechanical properties over a 12-month period under natural weathering [52] and decreases the water absorption and increases the hydrophobicity of the wood [30]. The application of a polymer matrix without the addition of nanomaterials is not sufficient to obtain a unique multifunctional consolidation solution that overcomes all of the wood’s disadvantages. The need to add nanomaterials was demonstrated in several stages of our study (e.g., in order to improve the mechanical (Figure 9) or antifungal properties (Figure 11)). Applying only the obtained PHBHV solution on the wood surface only led to an improvement in the water absorption properties, as can be seen in Figure 6, where similar values were obtained for all treated samples, the main reason being that the same polymer concentration (2%) was used for all of the applied treatments.

We hypothesized that the obtained PHBHV solutions containing NPs on the MWCNT surfaces would be the most suitable as consolidation treatment. ZnO, Ag and HAp nanoparticles were specifically selected for the MWCNT decoration, as literature studies suggest that these nanoparticles act as compatibilizers by incorporating into the gaps and spaces of MWCNTs, thus increasing the mechanical properties and hydrophobicity of the materials on which they are applied. By decorating the MWCNTs with nanoparticles, the disadvantages of the CNTs can be overcome; thus, the nanotubes are no longer inert, and their dispersion is improved. Thus, using simple blends of MWCNTs and nanoparticles could lead to agglomerated areas of CNTs and NPs, respectively, which is not desirable for wood treatment.

Furthermore, we believed that the treatment methods and nanocomposite concentrations would significantly enhance the properties of the wood. Our concern was related to the most important criterion, namely, that the obtained solutions should not change the natural color of the wood. In order to test these hypotheses, the treatments’ potential was investigated by a number of complex methods, such as consolidant retention, colorimetric parameters, water absorption, humidity, contact angle, mechanical tests, artificial aging and antifungal tests.

Experiments demonstrated that the application method strongly influenced the amount of consolidant deposited on the wood and the effectiveness of the protective treatment as a consequence. This was supported by the OM results, which showed a thicker layer of polymer on the surface of the wood treated by brushing. A possible explanation for the presence of the thicker layer of polymer on the wood surface could be related to the high solvent evaporation rate when the brushing method was used to coat the wood [53]. This may impede the penetration of the polymer into the wood microstructure, thus obtaining a thicker layer of polymer that, when applied to the wood surface, will protect the wooden material, providing a durable surface that helps to prevent damage and keeps it in good condition.

In colorimetric tests, low differences in color after the treatment application were observed at a lower nanomaterial concentration; conversely, higher values were obtained when a 0.4% concentration of nanocomposites was used. This suggests that the nanocomposite quantity in the solutions must be limited, and the acceptable nanocomposite concentrations in the solution are between 0.1 and 0.2%.

Water contact angle measurements indicated decreased wood wettability after polymeric treatments, which is a good surface characteristic for conservation and restoration applications in order to prevent moisture absorption. Regarding the results obtained from the hydrophobicity tests, the best results were obtained for samples treated by brushing with a 0.2% nanocomposite concentration. It can be concluded that in order to obtain better water protection, a thicker layer must be created on the wood surface. Additionally, the fastest reduction in the water contact angle occurred for wood samples treated by immersion, suggesting the ability of water to spread, which affects the quality of the final coating, thus decreasing the performance of the used consolidation solution.

The importance of the nanoparticles attached to the CNTs was confirmed by mechanical tests. It can be observed that the mechanical properties of wood treated with the MWCNTs + PHBHV solution were almost similar to those of the untreated wood. A possible explanation is related to the tendency of CNTs to form agglomerates in the solution, and this can affect the mechanical properties of the wood. When decorated CNTs were used, a slightly higher compressive strength compared with the untreated wood was observed (Figure 9). This observation was expected, judging from the fact that the presence of the nanoparticles located on the nanotube surface improves the CNT dispersion in the polymer solution. As expected, the presence of ZnO nanoparticles on MWCNTs improved the mechanical properties due to the chemical reaction formed between nanoparticles with the existing functional groups of individual CNTs [54].

Accelerated aging behavior, studied by UV radiation and by temperature variation exposure, demonstrated that the exterior environment has a significant impact on untreated wood. When the treatments were used, there was a significant slowdown in the discoloration of wood materials, especially when decorated nanotubes were present in the solution. This confirms that the presence of certain absorbent nanoparticles, such as ZnO NPs, on the surface of the nanotubes prevents the degradation of lignin under the action of UV light. It was reported that nanocomposites formed by the dispersion of inorganic nanoparticles in a polymer matrix improved abrasion, scratch resistance, heat resistance, toughness and stiffness. ZnO nanoparticles have the ability to offer UV protection to coatings and the underlying substrates while also being transparent in the visible spectrum [55]. The photostability and thermal stability of ZnO NPs provide advantages, such as being stable and non-migratory within a polymeric matrix, and thus potentially impart better effectiveness and longer service life. In comparison with other UV-protection candidates, ZnO is advantageous due to its protection over long periods and its broadband protection [56]. The performed experiments are equivalent to two years of aging in natural conditions, which suggests that the treatment based on MWCNTs_ZnO + PHBHV can be used for at least two years with the same good performance as on the first day. Aggressive environmental factors, such as UV degradation, affect many natural and synthetic polymers, which can lead to the fading of wood color, the loss of wood strength and a reduction in the water resistance of wood and can cause further biodegradation. For this reason, MWCNTs and nanoparticles were used in order to improve the UV resistance of the coating and to provide a high level of UV protection. Compared with other nanomaterials, CNTs and ZnO NPs greatly improve the efficiency of blocking ultraviolet rays.

Since any polymer can be affected by UV light, it is necessary to protect the wooden material by using a biodegradable polymer. For this reason, it is preferable to use a biodegradable polymer that does not affect the structure of the wood material and can present ‘reversibility by biodegradation’ for easy re-treatability, regain their property and improve durability.

Another essential parameter improved by the applied treatment on wood was the antifungal activity. The antifungal activity was improved with the decoration of the nanotubes, especially when nanoparticles with potent broad-spectrum antifungal activity, such as Ag and ZnO NPs, were used, thus confirming the importance of these agents in the solution.

Our goal was to obtain a new consolidation solution that covers most of the disadvantages of wooden materials; therefore, all of the results obtained in this study were accumulated in order to choose the best solution. With the complete analysis of the results, it can be concluded that the most effective treatment was MWCNTs_ZnO + PHBHV at a nanocomposite concentration of 0.2%, applied by brushing. Thus, the potential practical application of this solution will help to prevent many of the conditions that can cause the degradation and disintegration of wood over time. Compared with other treatments presented in the literature [57], our treatment avoids the dispersion problem, the nanocomposites being well dispersed in the obtained solution, because of the chemical reaction formed between the nanoparticles and individual MWCNTs. Another problem resolved by using this treatment is related to the preservation of the natural color of the wood [58], which was demonstrated above. Additionally, both the antimicrobial [26] and antifungal activities of the coating were confirmed in order to prevent the growth of bacteria, fungi and other microorganisms, compared with other studies [59].

By using this treatment, wood will be protected against mold and fungi, the most common causes of wood rot and decay, and will have enhanced mechanical properties and lower water absorption from the environment. Therefore, the coating will improve the durability of the wood material over time.

## 5. Conclusions

This study investigated the capacity of new solutions based on MWCNTs dispersed in PHBHV solution and decorated MWCNTs dispersed in PHBHV solution applied on wood pieces at various doses of nanocomposites. The consolidation potential after wood treatment was investigated by various methods in order to find the best consolidation solution applied by the appropriate method.

According to the obtained results, the highest consolidant retention was obtained by the brushing application. When using this method, a uniform layer was created on the wood surface that was able to ensure better protection of the wood material. According to colorimetric investigations, the 0.2% nanocomposite concentration was optimal and did not change the natural wood color. In addition, the wood pieces treated with MWCNTs_ZnO + PHBHV and MWCNTs_HAp + PHBHV solutions presented the lowest color change. A significant improvement of the treated samples compared to the control was observed in the water-based tests. The used treatments were able to protect the wood material, thus preventing the water penetration into the wood structure. The best results were obtained for the wood samples treated with the solutions based on 0.2 and 0.4% nanocomposite concentrations, applied by brushing and spraying.

Regarding the wood behavior with accelerated aging by UV radiation exposure, it was demonstrated that the control degraded faster, the total color difference being much higher compared to the treated samples. The wood pieces treated by brushing with solutions based on a 0.2% nanocomposites concentration presented greater protection against UV and temperature exposure, thus confirming better protection over time. Following the antifungal tests, higher growth inhibition was obtained for the wood samples treated with MWCNTs_ZnO + PHBHV and MWCNTs_Ag + PHBHV. Our results support the idea that coatings based on 0.2% MWCNTs_ZnO + PHBHV and 0.2% MWCNTs_Ag + PHBHV applied by brushing are able to protect wooden material against environmental factors, thus making them successful candidates for the development of efficient consolidants.

The untreated sound oak wood showed good compatibility with the proposed treatments, and treating the wood pieces provided major benefits from all points of view. Considering all of the results obtained from the performed characterization techniques, it can be concluded that the coatings based on 0.2% MWCNTs_ZnO + PHBHV, applied by brushing, present very intriguing properties that make them capable of future practical applications.

## Figures and Tables

**Figure 1 nanomaterials-12-01990-f001:**
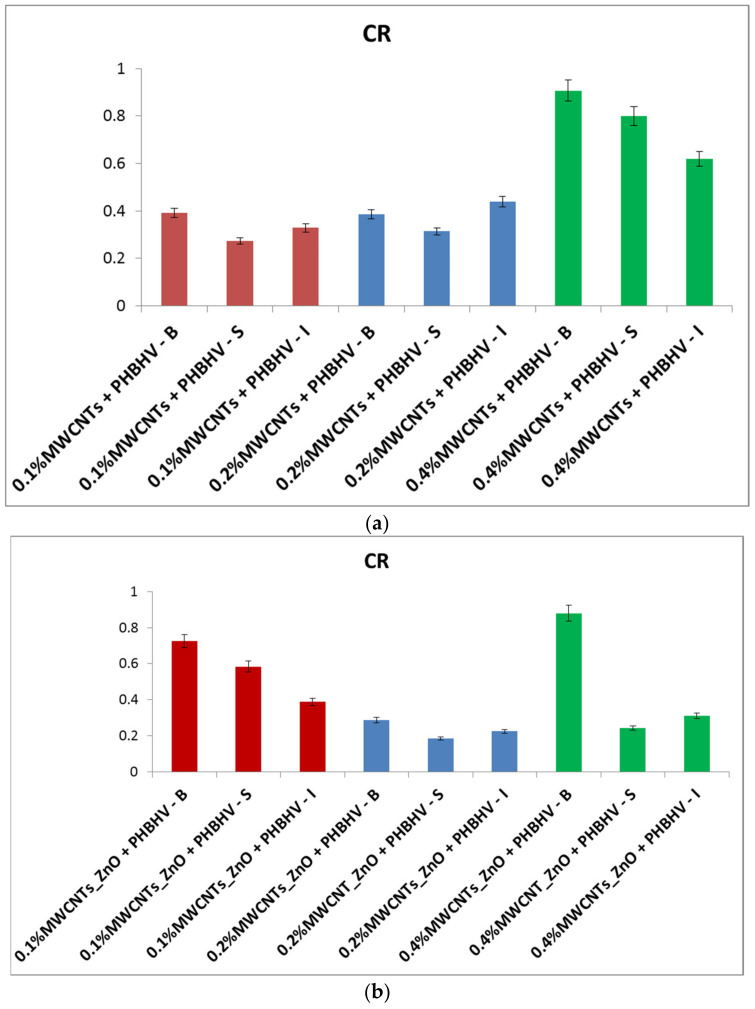
CR calculated for the treated samples: (**a**) MWCNTs + PHBHV set, (**b**) MWCNTs_ZnO + PHBHV set, (**c**) MWCNTs_HAp + PHBHV set and (**d**) MWCNTs_Ag + PHBHV set. The results are presented as the mean ± S.D. of three replicates.

**Figure 2 nanomaterials-12-01990-f002:**
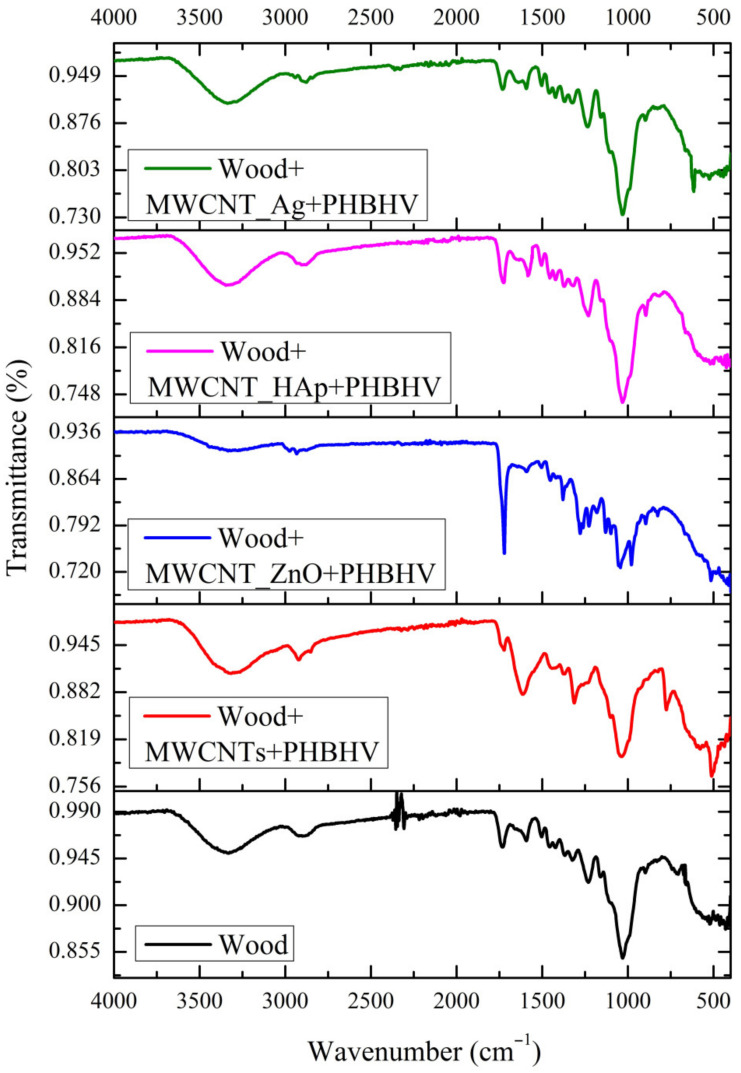
FTIR spectra for untreated and treated wood.

**Figure 3 nanomaterials-12-01990-f003:**
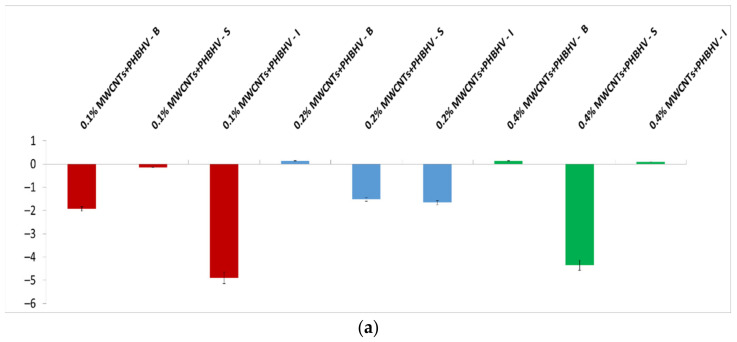
Calculated lightness difference (ΔL_x_): (**a**) MWCNTs + PHBHV set, (**b**) MWCNTs_ZnO + PHBHV set, (**c**) MWCNTs_HAp + PHBHV set and (**d**) MWCNTs_Ag + PHBHV set. The results are presented as the mean ± S.D. of three replicates.

**Figure 4 nanomaterials-12-01990-f004:**
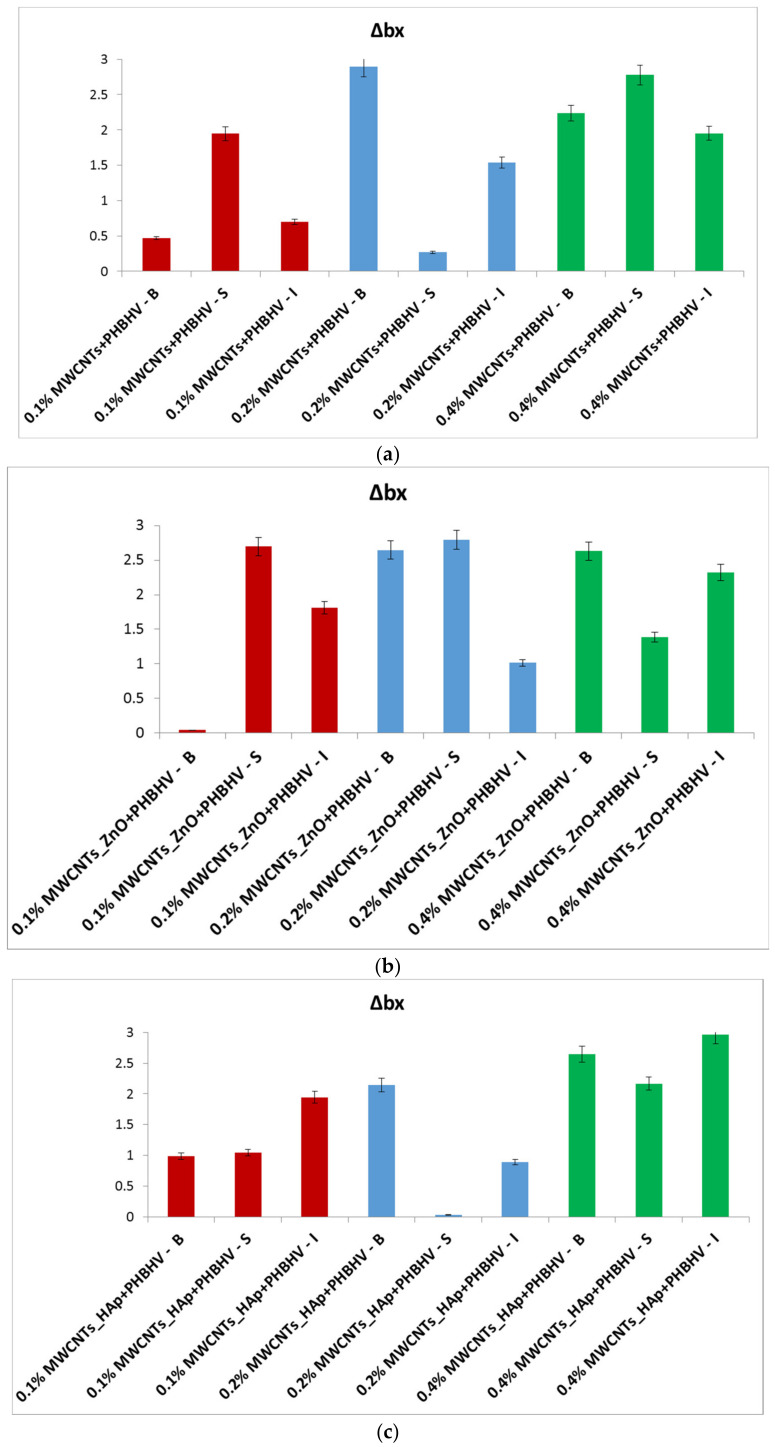
Calculated yellow-blue difference (Δb_x_): (**a**) MWCNTs + PHBHV set, (**b**) MWCNTs_ZnO + PHBHV set, (**c**) MWCNTs_HAp + PHBHV set and (**d**) MWCNTs_Ag + PHBHV set. The results are presented as the mean ± S.D. of three replicates.

**Figure 5 nanomaterials-12-01990-f005:**
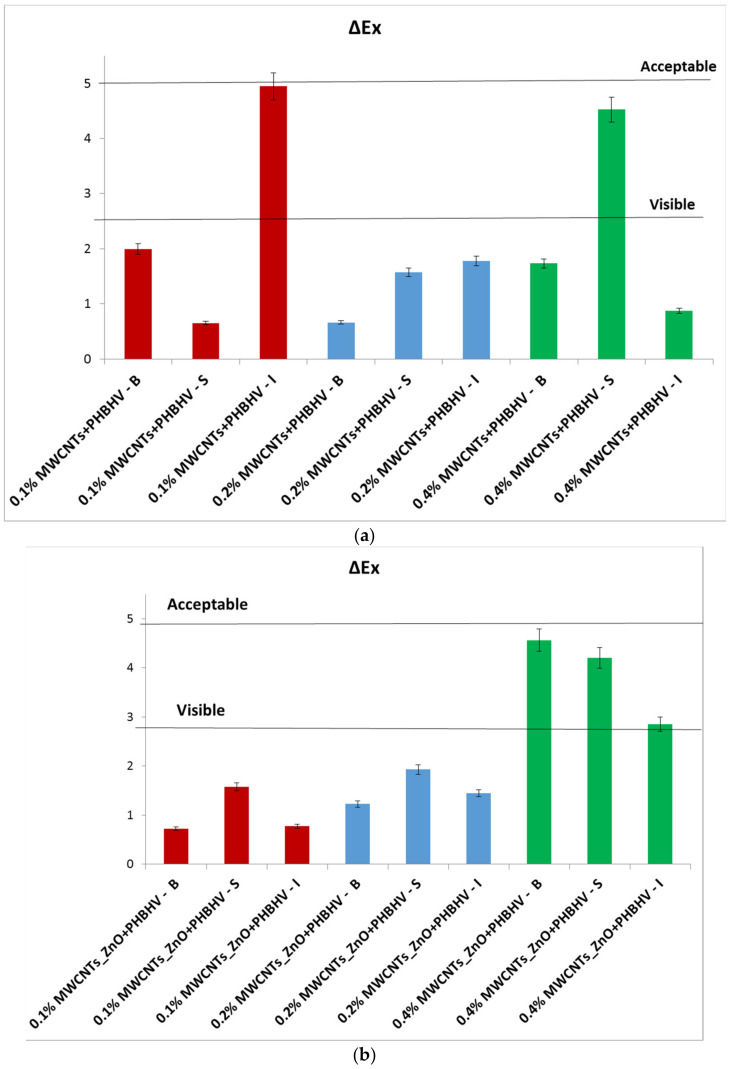
Total color difference calculated (ΔE_x_): (**a**) MWCNTs + PHBHV set, (**b**) MWCNTs_ZnO + PHBHV set, (**c**) MWCNTs_HAp + PHBHV set and (**d**) MWCNTs_Ag + PHBHV set. The results are presented as the mean ± S.D. of three replicates.

**Figure 6 nanomaterials-12-01990-f006:**
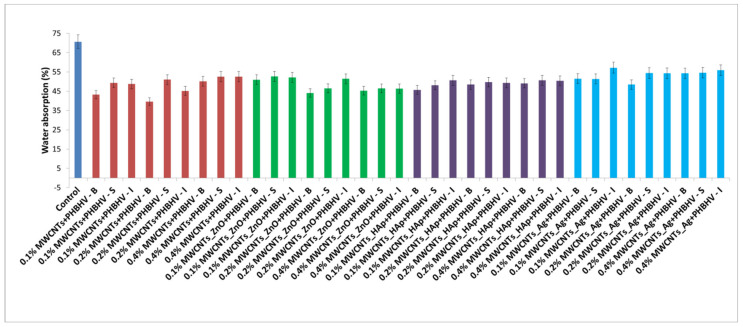
WA for untreated and treated samples.

**Figure 7 nanomaterials-12-01990-f007:**
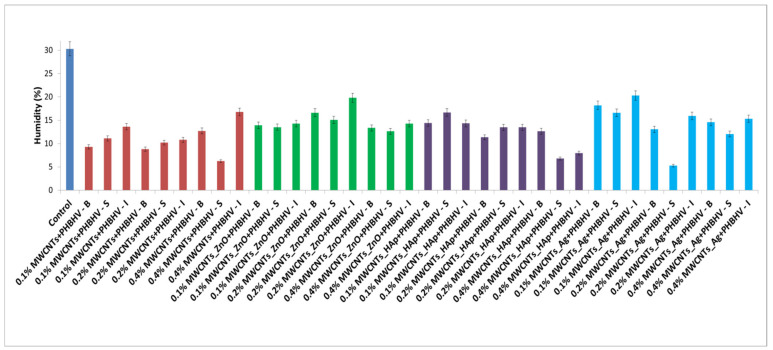
Obtained results from U test for untreated and treated samples.

**Figure 8 nanomaterials-12-01990-f008:**
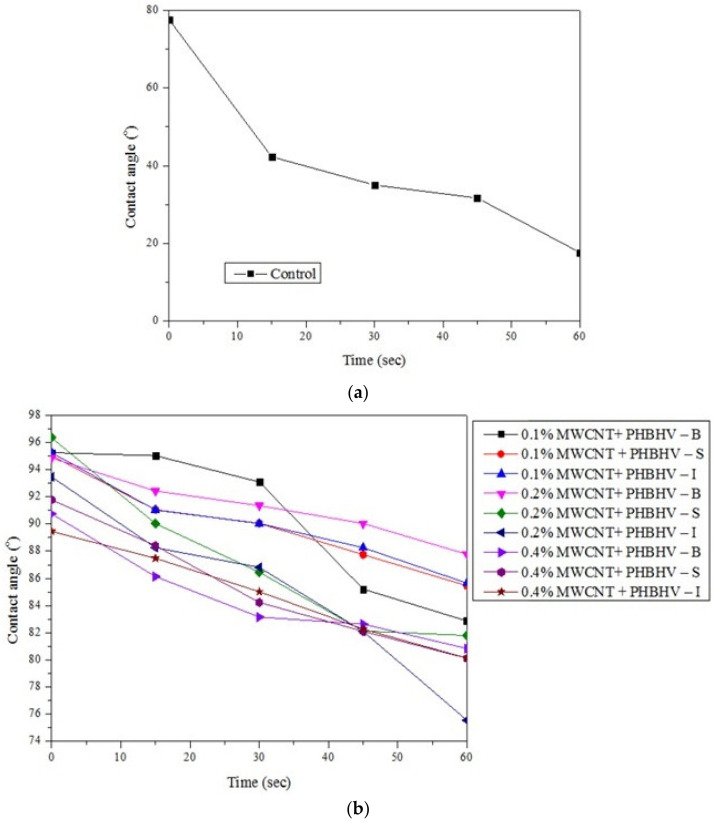
Calculated contact angles for untreated (**a**) and treated samples with (**b**) MWCNTs + PHBHV set, (**c**) MWCNTs_ZnO + PHBHV set, (**d**) MWCNTs_HAp + PHBHV set and (**e**) MWCNTs_Ag + PHBHV set.

**Figure 9 nanomaterials-12-01990-f009:**
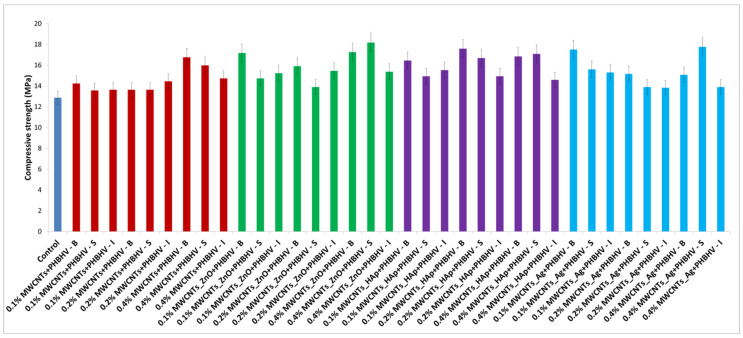
Compressive strength of untreated and treated samples.

**Figure 10 nanomaterials-12-01990-f010:**
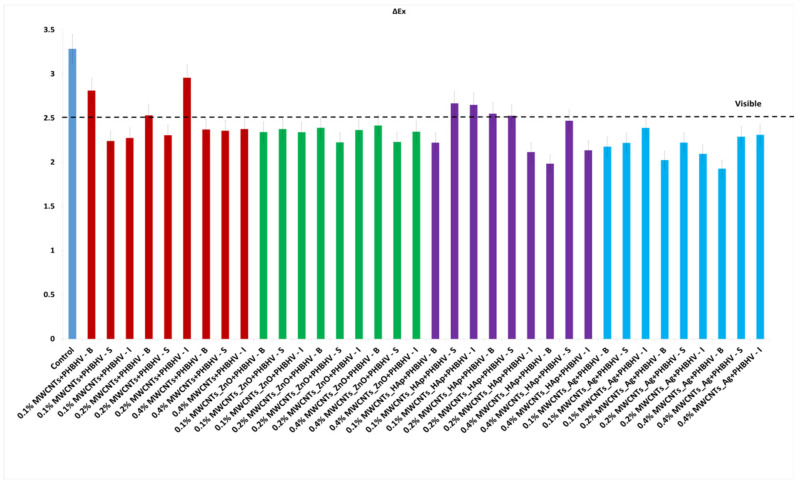
Total color difference calculated after artificial aging test based on temperature variation exposure.

**Figure 11 nanomaterials-12-01990-f011:**
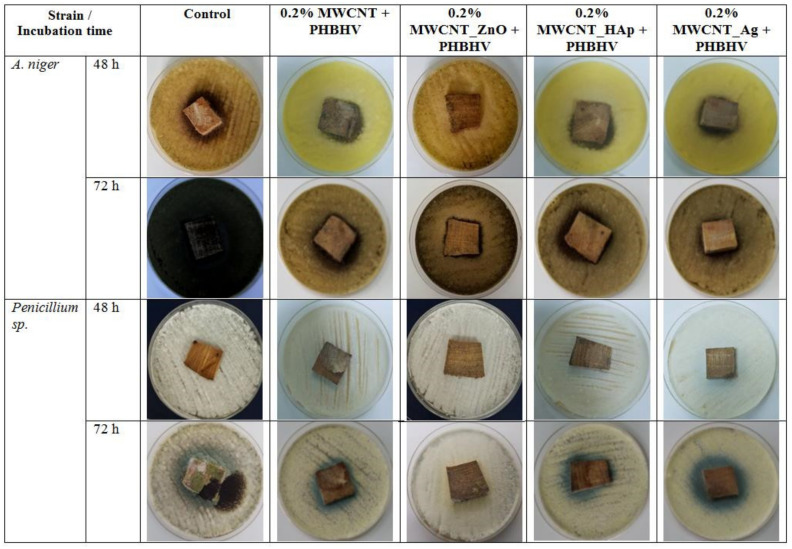
Images after antifungal test for untreated and treated wood samples.

**Table 1 nanomaterials-12-01990-t001:** Thickness of the untreated and treated wood pieces by brushing, immersion and spraying obtained by optical microscopy.

Sample	Central Zone (10×)	End Zone (4×)
Control	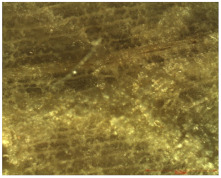	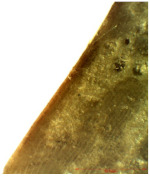
0.1% MWCNT+ PHBHV—brushing	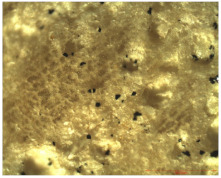	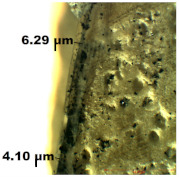
0.1% MWCNT+ PHBHV—immersion	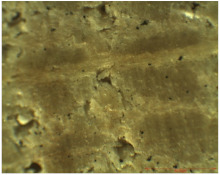	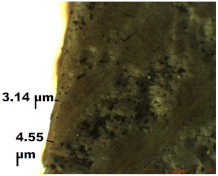
0.1% MWCNT+ PHBHV—spraying	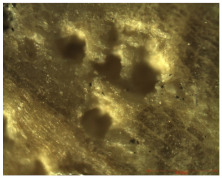	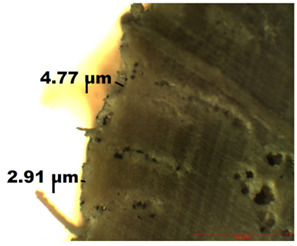

**Table 2 nanomaterials-12-01990-t002:** Composition of untreated and untreated samples investigated by WDXRF.

Compound	Wood (%)	Wood + MWCNTs + PHBHV (%)	Wood+ MWCNTs_ZnO + PHBHV (%)	Wood+ MWCNTs_HAp + PHBHV (%)	Wood + MWCNT_Ag + PHBHV (%)
SiO_2_	5.1821	25.7321	7.4745	18.4805	7.7196
SO_3_	5.8244	4.0752	3.0447	3.4126	4.1864
Cl	1.2235	7.6076	1.4374	1.7123	3.5031
K_2_O	14.6479	18.3917	5.6448	11.7196	23.5348
CaO	59.4586	28.5891	68.4308	39.4028	49.5594
Fe_2_O_3_	6.9201	1.4846	-	2.287	3.7029
P_2_O_5_	4.7242	3.3406	1.1225	22.5611	3.0425
Al_2_O_3_	1.0652	7.3128	7.1676	-	-
MnO	0.5368	1.6982	-	-	2.5784
MgO	0.4172	1.7681	3.7201	0.4241	1.8244
ZnO	-	-	1.9576	-	-
Ag_2_O	-	-	-	-	0.3485

**Table 3 nanomaterials-12-01990-t003:** ANOVA, color change with consolidation.

Dependent Variable: ΔE_x_ (After Consolidation)							
Variables		Source of Variation	Sum of Squares	Df	Mean Square	F	*p*-Value
Type of treatment	MWCNT+ PHBHV	Concentration	1.9256	2	0.9628	0.3514	0.7172
		Method	9.5254	3	3.1751	1.1590	0.3996
	MWCNT_ZnO+ PHBHV	Concentration	11.0994	2	5.5497	8.3345	0.0185
		Method	9.4055	3	3.1351	4.7084	0.0410
	MWCNT_HAp+ PHBHV	Concentration	7.6432	2	3.8216	7.4622	0.0235
		Method	4.9421	3	1.6473	4.7166	0.0439
	MWCNT_Ag+ PHBHV	Concentration	2.8366	2	1.4182	0.9158	0.4496
		Method	8.6371	3	2.8790	1.8592	0.2372
Type of treatment application	Brushing	Brushing method vs. concentration	14.7134	1	14.7134	19.8295	0.0001
	Spraying	Spraying method vs. concentration	25.2806	1	25.2806	16.8983	0.0005
	Immersion	Immersion method vs. concentration	25.0684	1	25.0684	16.5376	0.0005
Nanocomposite concentration	0.1%	Concentration type vs. methods	10.0715	1	10.0715	12.1486	0.0020
	0.2%	Concentration type vs. methods	8.8493	1	8.8493	20.7264	0.0001
	0.4%	Concentration type vs. methods	19.5577	1	19.5577	15.7231	0.0047

**Table 4 nanomaterials-12-01990-t004:** ANOVA, mechanical improvement achieved with consolidation.

Dependent Variable: Compressive Strength (After Consolidation)						
Type of Treatment	Source of Variation	Sum of Squares	Df	Mean Square	F	*p*-Value
MWCNT+ PHBHV	Concentration	4.3569	2	2.1784	1.8139	0.2420
	Method	556.0621	3	185.3540	154.3387	4.557 × 10^−6^
	Error	7.2057	6	1.2009		
MWCNT_ZnO+ PHBHV	Concentration	6.2816	2	3.1408	5.1534	0.0498
	Method	459.2200	3	153.0733	251.1636	1.075 × 10^−6^
	Error	3.6567	6	0.6094		
MWCNT_HAp+ PHBHV	Concentration	0.8043	2	0.4021	0.8123	0.4872
	Method	569.7249	3	189.9083	383.6067	3.046 × 10^−7^
	Error	2.9703	6	0.4950		
MWCNT_Ag+ PHBHV	Concentration	3.9594	2	1.9797	1.3635	0.3249
	Method	517.4135	3	172.4711	118.7925	9.867 × 10^−6^
	Error	8.7112	6	1.4518		

**Table 5 nanomaterials-12-01990-t005:** Total color difference calculated after artificial aging test by UV radiation exposure (ΔEx/Exposure time).

Exposure Time (h)	ΔEx
100	
	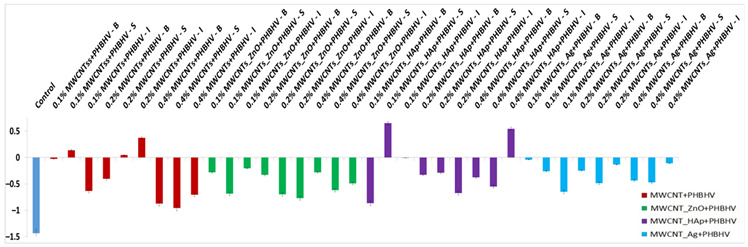
200	
	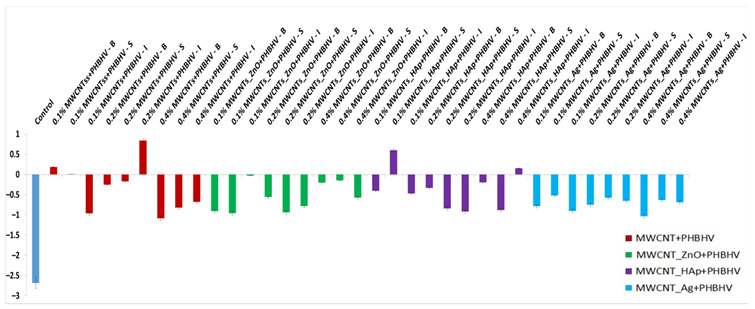
300	
	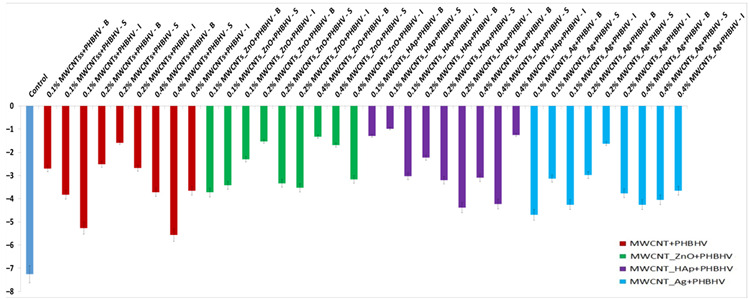
400	
	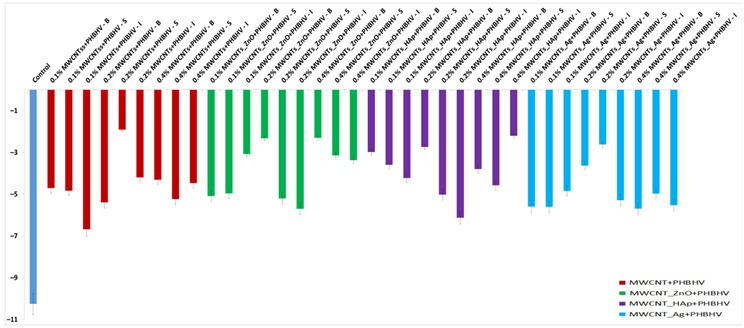

## Data Availability

Data are contained within the article.

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
