# Peer review of "Wood Surface Modification with Hybrid Materials Based on Multi-Walled Carbon Nanotubes"

_nanomaterials, 2022, doi:10.3390/nano12121990_

Round 1
Reviewer 1 Report
PHBHV was used by the Authors already in previous works and this article is a report on the continuation of this research. Nevertheless, the use of PHBHV as a component of wood surface protection needs better justification as its biodegradability, which I'm my opinion is a disadvantage in the case of materials intended to protect surfaces for a long, indefinite period of time. Please explain.
If the research were to be used practically, it is necessary to compare the obtained results to the current state of knowledge and practice in the field of wood surface protection, because the appropriate technologies have been used for a long time. The competitiveness of the new technology should be well demonstrated. I recommend Bi, W., Li, H., Hui, D., Gaff, M., Lorenzo, R., Corbi, I., Corbi, O., Ashraf, M., 2021. Effects of chemical modification and nanotechnology on wood properties. Nanotechnology Reviews 10, 978–1008 .. doi: 10.1515 / ntrev-2021-0065 and Modification of Wood Surfaces ". A special issue of Polymers (ISSN 2073-4360).
Author Response
Many thanks for your valuable comments and recommendation, which helped us to greatly improve our paper. We made important corrections evidenced by track changes and tried to answer all your requests as good as possible.
According to your recommendation:
- The presence of PHBHV in the solution was justified in the discussion section
- Our product with the best results (MWCNTs_ZnO+PHBHV) was compared with the obtained results to the current state of knowledge and practice in the field of wood surface protection.
- The indicated reference has been included in the paper and other papers were included.
Reviewer 2 Report
The Article: “Wood Surface Modification with Hybrid Materials Based on Multi-Walled Carbon Nanotubes” by Madalina Elena David et al. describes the use of new treatments based on multi-walled carbon nanotubes (MWCNTs), decorated with zinc oxide or with hydroxyapatite, or with silver nanoparticles dispersed in PHBHV solution for increasing the sound oak wood properties.
In the paper the results part is presented but the discussion part is poor What effect does the application of the polymer matrix poly(3-hydroxybutyrate-co-3-hydroxyvalerate) without nanotubes have on the wood?
Did the authors test the simple blend between MWCNTs and nanoparticles? What is the difference from using MWCNTs decorated or only blended with nanoparticles? The synthesis of MWCNTs decorated is expensive in terms of time and costs and must therefore be justified.
What is the effect of the polymer matrix with only nanoparticles without nanotubes?
The authors have described the effect on wood of decorated MWCNTs but they also have to try to understand why and this has been done only on the paragraph related to the Antifungal activity.
For this reason, without the addition of a part of discussion in my opinion the work cannot be published in a journal such as NANOMATERIALS.
Here are some other suggestions
Figures :
Fig 1:The caption should be explicative, what is a) b),c) and d)? They are not mentioned in the text .
What about error bars? They are never mentioned in the main text or in the captions.
Table 1: It seem to be a figure and not a table. The numbers in the right part of the figure are difficult to read please enlarge. The caption should be more explicative and all the data reported in the figure should be discussed in the manuscript. I suppose the Authors reported the thickness of the coating but no mention are done in the main text or in the captions.
Fig 3: to improve the readability I suggest to translate the description in the upper part of the graph. As for Fig 1 the caption should be explicative, what is a) b),c) and d)? They are not mentioned in the text and in the caption.
Fig 4 The caption should be explicative, what is a) b),c) and d)? They are not mentioned in the text and in the caption.
Fig 5 The caption should be explicative, what is a) b),c) and d)? They are not mentioned in the text and in the caption.
Table 5 is difficult to read. I suggest to turn the captions in the upper part of the graph
What kind of analysis after ageing?
Could Artificial aging by UV radiation exposure have an impact not only on the colour but also on other parameters? Usually UV degrades the organic polymeric host and many properties like contact angle or composition (IR WDXRF analysis) or Water absorption could change. Please discuss.
Authors suggested that the applied method is an important factor in improving the wood properties. What is the role of the method and the role of the thickness of the coting? Please discuss
Self-Citation 8 on 50 equal to 16%: OK .
Author Response
Many thanks for your valuable comments and recommendation, which helped us to greatly improve our paper. We made important corrections evidenced by track changes and tried to answer all your requests as good as possible.
In the paper the results part is presented but the discussion part is poor What effect does the application of the polymer matrix poly(3-hydroxybutyrate-co-3-hydroxyvalerate) without nanotubes have on the wood?
- Answer: The effect of the application of PHBHV solution on the wood was explained in the discussion part.
Did the authors test the simple blend between MWCNTs and nanoparticles? What is the difference from using MWCNTs decorated or only blended with nanoparticles? The synthesis of MWCNTs decorated is expensive in terms of time and costs and must therefore be justified.
- Answer: The difference from using MWCNTs decorated or only blended with nanoparticles was explained in the discussion part. The synthesis of MWCNTs decorated is not expensive, following our synthesis methods, previous published [Hybrid Materials Based on Multi-Walled Carbon Nanotubes and Nanoparticles with Antimicrobial Properties].
What is the effect of the polymer matrix with only nanoparticles without nanotubes?
- Answer: The effect of the polymer matrix with only nanoparticles without nanotubes has not been studied, but it is part of our concerns and will be studied in a future work.
The authors have described the effect on wood of decorated MWCNTs but they also have to try to understand why and this has been done only on the paragraph related to the Antifungal activity.
- Answer: The detailed discussion part was added.
For this reason, without the addition of a part of discussion in my opinion the work cannot be published in a journal such as NANOMATERIALS.
- Answer: We improved a lot the Discussion part and we tried to reach the Results part
Here are some other suggestions
Figures :
Fig 1:The caption should be explicative, what is a) b),c) and d)? They are not mentioned in the text. What about error bars? They are never mentioned in the main text or in the captions.
- Fig 1: The caption was explained and the error bars were mentioned
Table 1: It seem to be a figure and not a table. The numbers in the right part of the figure are difficult to read please enlarge. The caption should be more explicative and all the data reported in the figure should be discussed in the manuscript. I suppose the Authors reported the thickness of the coating but no mention are done in the main text or in the captions.
- Table 1: The numbers in the figures were enlarged, the caption was explained, and the thickness was discussed in the discussion part.
Fig 3: to improve the readability I suggest to translate the description in the upper part of the graph. As for Fig 1 the caption should be explicative, what is a) b),c) and d)? They are not mentioned in the text and in the caption.
- Fig 3 and caption was modified according to your suggestions.
Fig 4 The caption should be explicative, what is a) b),c) and d)? They are not mentioned in the text and in the caption.
- Fig 4 The caption was explained
Fig 5 The caption should be explicative, what is a) b),c) and d)? They are not mentioned in the text and in the caption.
- Fig 5 The caption was explained
Table 5 is difficult to read. I suggest to turn the captions in the upper part of the graph
- Table 5 was modified according to your suggestions.
What kind of analysis after ageing? Could Artificial aging by UV radiation exposure have an impact not only on the colour but also on other parameters? Usually UV degrades the organic polymeric host and many properties like contact angle or composition (IR WDXRF analysis) or Water absorption could change. Please discuss.
- The effect of the artificial aging by UV radiation exposure was discussed in proper details in the discussion part.
Authors suggested that the applied method is an important factor in improving the wood properties. What is the role of the method and the role of the thickness of the coting? Please discuss
- The role of the method and the role of the thickness of the coting was discussed in the discussion part.
Self-Citation 8 on 50 equal to 16%: OK .
Round 2
Reviewer 2 Report
The work of Madalina Elena David et al. has been significantly improved however in some parts the discussion of the results is highly speculative and the hypotheses must be confirmed by experimental data or supported by literature references. For this reason I suggest a further revision. Here are some suggestions.
There are some typos especially in the added text part, I suggest to read it carefully.
Lines 565-566 the authors reported : “..we selected PHBHV as dispersing solution…” PHBHV is a polymeric host not a solution.
Line 570 ref 51 The cited article is on stone , what about wood? The functional groups on the surface, the porosity and roughness of wood are very different from stone. Please discuss.
Line 580 “… We hypothesis…” please correct in we hypothesize.
Lines 595-600 This part is high speculative I suggest adding references and/or experimental results.
Lines 609-610 The sentence is not clear please rewrite.
Line 629 “…explanation is related by the…” please use “to” instead of “by”.
Lines 630-632 please add a reference or an experimental result supporting your affirmation
Lines 632 the Authors wrote : “This observation was expected by judging from the fact that the presence of the nanoparticles improved the dispersion of CNTs in the polymer solution”. This is a crucial point for this reason I suggest adding the experimental data that support this claim.
Lines 641-642 the Authors wrote : “the presence of certain absorbent nanoparticles, like ZnO NPs on the surface of the nanotubes prevents the degradation…”. Please discuss "why certain absorbent" prevent degradation. What is the role of zinc oxide? And what about “certain adsorbent?”
Line 651 ref 54 In the proposed reference is reported a study on MWCNT/amine-cured epoxy nanocomposite irradiated with high intensity ultraviolet (UV) light, the effect of UV exposure on the surface accumulation and potential release of MWCNTs, and possible mechanisms for the release resistance of the MWCNT .
The materials of this paper, polymeric matrix PHBHV and Zinc oxide and substrate are quite different. Please discuss.
Lines 672-673 This sentence is not clear please rewrite
The conclusion must be improved at the moment they are a list of experimental results.
Author Response
Many thanks for your valuable comments which greatly improved our paper. All your suggestions were taken into account and we have tried to respond as best we can.
Lines 565-566 - were corrected
Line 570 ref 51 – more results sustained by references were added
Line 580 “… We hypothesis…” – was corrected
Lines 595-600 - We think that this part is already sustained by both the results and statistical analysis presented in chapter 3
Lines 609-610 – it was rewrite
Line 629 – it was modified
Lines 630-632 – Figure 9 was mentioned
Lines 632 the Authors wrote : “This observation was expected by judging from the fact that the presence of the nanoparticles improved the dispersion of CNTs in the polymer solution”. - This was already discussed in the first part of the Chapter 4
Lines 641-642 the Authors wrote : “the presence of certain absorbent nanoparticles, like ZnO NPs on the surface of the nanotubes prevents the degradation…”. - The role of zinc oxide was discussed
Line 651 ref 54 – it was deleted
Lines 672-673 – it was rewrite
The conclusion was improved and modified